# Provable Defense Against Clustering Attacks on 3D Point Clouds

**Dishanika Denipitiyage**[1], **Thalaiyasingam Ajanthan**[2], **Parameswaran Kamalaruban**[3], and **Adrian Weller**[3,4]

[1] Department of Electronic and Telecommunication Engineering, University of Moratuwa
[2] The Australian National University, Amazon Australia
[3] The Alan Turing Institute
[4] University of Cambridge

dewni.denipitiya@gmail.com, thalaiyasingam.ajanthan@anu.edu.au, kparameswaran@turing.ac.uk, aw665@cam.ac.uk

## Abstract

Lately, the literature on adversarial robustness spans from images to other domains such as point clouds. In this work, we consider clustering attacks on 3D point clouds and devise a provable defense mechanism to counter them. Specifically, we adopt a randomized smoothing strategy for 3D point clouds and derive a robustness certificate based on the cluster radius rather than the number of adversarial points. Our experiments on ModelNet40 and ScanObjectNN datasets using the Point-Net classifier demonstrate the effectiveness of our defense mechanism against targeted and untargeted clustering attacks with a large number of adversarial points.

## 1 Introduction

Adversarial robustness is an important research topic both in terms of understanding modern neural networks and in safety-critical applications (Akhtar and Mian 2018). Recently, this topic is becoming increasingly popular in domains outside of images, including point clouds.

The recent literature on adversarial robustness on point clouds mainly focuses on point perturbation/addition/deletion attacks (Liu, Jia, and Gong 2021). However, despite their practical importance in applications such as self-driving cars (Cao et al. 2019; Xiang et al. 2021), the stronger clustering attacks are less studied (Xiang, Qi, and Li 2019).

In this work, we consider clustering attacks on 3D point clouds and devise a provable defense mechanism to counter them. Our idea is to adopt a (de)-randomized smoothing strategy (Levine and Feizi 2020) for 3D point clouds. Specifically, we divide the 3D space into equally sized voxels, and learn a classifier on points contained in the randomly subsampled voxels (see Figure 1). We show that such a classifier is robust to a large number of adversarial points as long as they are concentrated on a small set of clusters. In short, we derive a robustness certificate that is based on the attack cluster radius rather than the number of adversarial points.

We evaluate our approach on a synthetic ModelNet40 dataset (Wu et al. 2015) and a real ScanObjectNN dataset (Dai et al. 2017). Our experiments demonstrate that our defense mechanism is significantly better than the comparable baselines against both targeted and untargeted clustering attacks when the number of adversarial points grows.

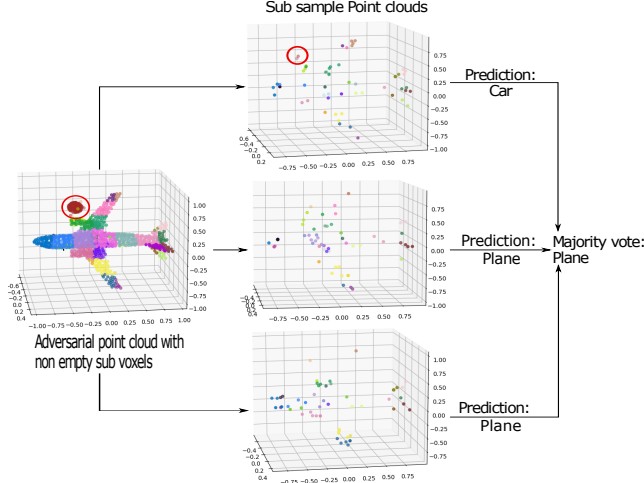

Sub sample Point clouds

Adversarial point cloud with non empty sub voxels

Prediction: Car

Prediction: Plane

Prediction: Plane

Majority vote: Plane

Figure 1: An illustration of our randomized subsampling strategy. Three subsampled point clouds are created, and one contains the adversarial points. The red circle covers the adversarial points. Our defense method predicts the correct label for the adversarial point cloud via majority vote.

## 2 Related Work

Existing adversarial attacks on 3D point clouds can be categorized into three categories based on the attacker's capability. The first type is point perturbation which is adopted from image-level adversarial attacks. Xiang et al. (Xiang, Qi, and Li 2019) used an optimization-based strategy to generate an adversarial noise. The objective of this attack is to find a minimal perturbation sample that can make the classifier classify incorrectly. The perturbation measurements include $\ell_2$-norm, Hausdorff distance, and Chamfer distance. Zhou et al. (Zhou et al. 2020) proposed a point cloud generation-based targeted attack, which learns how to deform the point cloud with minimal perturbation and then mislead the classifier into a specific label.

The second type is point deletion-based attacks. Zheng et al. (Zheng et al. 2019) proposed a gradient-based salience and dropping points attack with the lowest salience score. Yang et al. (Yang et al. 2019) proposed a method that is mainly focused on PointNet architecture that utilizes the crit-

ical point property. Critical points remain active after using max pooling, i.e., these points are important when determining the object category. Therefore missing critical points are more likely to change the prediction output. This intuition was used by Ying et al. (Yang et al. 2019) by iteratively removing critical points to create an untargeted attack.

The third type of attack is point addition; this can be adding random points, a cluster, or an object to the original point cloud. Xiang et al. (Xiang, Qi, and Li 2019) places a set of independent points or a limited number of point clusters and optimizes the cluster location and the shape of the clusters. Yang et al. (Yang et al. 2019) proposed a point-wise gradient-based point addition attack that updates the attacked points without changing the original points.

Several empirical defenses have been proposed to mitigate these attacks. These defenses are mainly focused on detecting attacks or training more robust classifiers. Adversarial training (Liu, Yu, and Su 2019) is one of the most effective methods to improve the robustness of the model by augmenting the training set with adversarial samples. DUP-Net (Zhou et al. 2019) proposed a statistical outlier removal denoiser and an upsampling network as a pre-processing strategy. Denoiser removes noise patterns as a non-deferential layer while the upsampling layer generates dense point clouds. IF-Dense (Wu et al. 2020) network proposed a solution to both point distribution changes due to the point perturbation and surface distortion. They restore the original point cloud using geometric and distribution-aware constraints. However, these networks do not provide any theoretically proven robustness guarantee.

Randomized smoothing (Cohen, Rosenfeld, and Kolter 2019) has achieved $\ell_2$-norm based certified robustness on ImageNet (for 2D classification problem), thus overcoming the limitations in the existing defense mechanisms against adversarial attacks. In general, this method constructs a new smoothed classifier $g$ from the base classifier $F$ where $g$ predicts the class which is more likely to be returned by the $F$ under the Gaussian noise perturbation of input. Point-Guard (Liu, Jia, and Gong 2021) is the first defense (for 3D point cloud classification problem) that provides provable robustness guarantees against adversarially modified, added, and/or deleted points. However, the certified accuracy of PointGuard is reduced when the number of adversarial points is increased. Thus, PointGuard is not effective under (highly dense) cluster attacks. In this work, we propose a provable defense mechanism against these clustering attacks.

# 3 Proposed Method

## 3.1 Preliminaries

**Point Cloud Classification** A point cloud is an unordered set of 3D coordinates which are sampled from object surfaces. We define a point cloud $P$ of size $n$ as $P = \left\{ P_i \mid P_i \in \mathbb{R}^3, i \in \{1, \ldots, n\} \right\}$, where each $P_i = (x, y, z)^\top$ is a point in 3D space. Let $F$ denote a point-based classifier that maps an input point cloud $P \in \mathbb{R}^{n \times 3}$ to its corresponding class label $y \in \{1, 2, \ldots, c\}$. Also, let $F_i(P)$ denote the probability that the point cloud $P$ is classified into the $i$-th class. Ideally, if $i^*$ is the true class label of the point cloud $P$, then $i^* = \arg\max_i F_i(P)$. Many deep learning

classifiers (e.g., (Qi et al. 2017a,b; Li et al. 2018; Wang et al. 2019)) have been proposed for point cloud classification. In this work, we mainly consider the PointNet (Qi et al. 2017a) classifier.

**Distance Measures** Let $D : \mathbb{R}^{n \times 3} \times \mathbb{R}^{n' \times 3} \to \mathbb{R}$ be a distance metric, i.e., $D(P, P')$ is some distance between two point clouds $P$ and $P'$. Below, we define a few distance metrics required for our purpose:

*(i) $\ell_p$-norm:* For the original point cloud $P$ and corresponding adversarial cloud $P'$, the $\ell_p$-norm (for $p \geq 1$) distance between $P$ and $P'$ is defined as:

$$D_{\ell_p}(P, P') = \frac{1}{n} \sum_{i=1}^{n} \|P_i - P_i'\|_p ,$$

where $P_i$ is the $i$-th point in $P$, and $P_i'$ is its corresponding point in $P'$.

*(ii) Chamfer measurement:* For the original point cloud $P$ and its adversarial counterpart $P'$, we define Chamfer Measurement (Fan, Su, and Guibas 2017) as:

$$D_{\mathrm{C}}(P, P') = \frac{1}{|P'|} \sum_{y \in P'} \min_{x \in P} \|x - y\|_2^2 ,$$

where $|P'|$ denotes the number of points in $P'$.

*(iii) Farthest distance:* We define farthest distance of a point cloud $P$ as (Xiang, Qi, and Li 2019):

$$D_{\mathrm{far}}(P) = \max_{x, y \in P} \|x - y\|_2 .$$

*(iv) Average distance:* We define average distance of a point cloud $P$ as:

$$D_{\mathrm{avg}}(P) = \frac{1}{n^n} \sum_{x, y \in P} \|x - y\|_2 .$$

Here we abuse the notation a little to use $D$ to denote both mappings $\mathbb{R}^{n \times 3} \times \mathbb{R}^{n' \times 3} \to \mathbb{R}$ and $\mathbb{R}^{n \times 3} \to \mathbb{R}$.

## 3.2 Clustering Attack

In this work, we consider the clustering attacks in point clouds, where an adversary adds a limited number ($m$) of adversarial shapes, as either generic primitive shapes such as balls or meaningful shapes such as small airplane models. In particular, we consider the targeted clustering attack model from (Xiang, Qi, and Li 2019) and untargeted clustering attack model from (Kim et al. 2021). These works showed that PointNet (Qi et al. 2017a) can be fooled by adding a limited number of synthesized point clusters with meaningful shapes. The number of clusters added is hard bounded to 1-3 in our experiments.

*(i) Targeted attack model:* Let $P'$ be an adversarial point cloud generated from the original point cloud $P$. The goal of the attack is to mislead the classifier $F$ to classify $P'$ as a selected target class. Let $t' \in \{1, 2, \ldots, c\}$ be the malicious target class of the adversary. The attack problem is formulated as follows:

$$\min_{P'} D(P, P'), \quad \text{s.t} \quad \arg\max_{i \in \{1, 2, \ldots, c\}} F_i(P) = t' .$$

The term, $D(P, P')$ constrains the perceptibility of the adversarial point cloud $P'$ w.r.t. the original point cloud $P$. Since directly solving the above constrained optimization problem is difficult, we reformulate it into an unconstrained optimisation problem using a Lagrange multiplier-like form as:

$$\min_{P'} \ h(P') + \lambda \cdot \mathcal{D}(P, P') ,$$

Here, $h(P') = \max\{0, \max_{i \neq t'} F_i(P') - F_{t'}(P')\}$ is the adversarial loss function whose output measures the possibility of a successful attack. By optimizing over this equation, we aim to search for adversarial examples with least 3D perturbation.

*(ii) Untargeted attack model:* In the untargeted attack model, the attacker aims to find an adversarial point cloud $P'$ as follows:

$$\min_{P'} \ D(P, P'), \text{ s.t } F_{i^*}(P) \neq \underset{i' \in \{1,2,\dots,c\}}{\arg\max} \ F_{i'}(P') ,$$

where $i^*$ is the true class label. The constraint $F_{i^*}(P) \neq \arg\max_{i'} F_{i'}(P')$ ensures the generated point cloud $P'$ can fool the network $F$, i.e., $F$ would not classify $P$ and $P'$ into the same class. Similar to the target attack model case, we reformulate above problem as follows:

$$\min_{P'} \ h(P') + \lambda \cdot \mathcal{D}(P, P') ,$$

where, $h(P') = \max\{0, F_{i^*}(P') - \max_{i' \neq i^*} F_{i'}(P')\}$ and $i^*$ is the true class label of $P$.

For the addition of adversarial shapes, we can use either generic primitive shapes (adversarial clusters) such as balls or meaningful shapes (adversarial objects) such as small airplane models.

*(i) Adversarial clusters:* Here, we aim to minimize the radius of the generated cluster so that the attack is a concentrated small cluster attached to the original object. In addition, we also encourage the cluster to be close to the object surface. These requirements are captured by the following distance metric:

$$D(P, P') = \sum_{i=1}^{m} \big\{ \mu_1 \cdot D_{\text{far}}(P^{(i)}) + \mu_2 \cdot D_{\text{avg}}(P^{(i)})$$
$$+ \mu_3 \cdot D_{\text{C}}(P, P^{(i)}) \big\},$$

where $P$ is the original object, $P^{(i)}$ is the $i$-th adversarial point cluster, $P' = P \cup P^{(1)} \cup P^{(2)} \cup \dots \cup P^{(m)}$, $m$ is the number of adversarial clusters, and $\mu_1, \mu_2, \mu_3 > 0$ balance the different terms.

*(ii) Adversarial objects:* Here, we start from some meaningful objects like small airplanes, slightly modify them, and place them in the appropriate adversarial positions. We consider the following distance metric to fit this attack setting:

$$D(P, P') = \sum_{i=1}^{m} \big\{ \mu_1 \cdot D_{\ell_2}(P^{(i,\text{org})}, P^{(i)})$$
$$+ \mu_2 \cdot D_{\text{C}}(P, P^{(i)}) \big\},$$

where $P$ is the original object, $P^{(i)}$ is the $i$-th adversarial point cluster, $P^{(i,\text{org})}$ is the $i$-th real-world point cluster, $m$ is number of adversarial clusters, and $\mu_1, \mu_2 > 0$.

## 3.3 Provable Defense

Here, we formalize our defense mechanism that is closely related to the certified defense against patch attacks for 2D image classification (Levine and Feizi 2020) and certified defense against point addition/deletion/perturbation attacks for 3D point cloud classification (Liu, Jia, and Gong 2021).

Consider a sufficiently large voxel of dimension $L_x \times L_y \times L_z$ that contains the input point cloud $P \in \mathbb{R}^{n \times 3}$. First, we block-partition this large voxel using non-overlapping small voxels of dimension $l_x \times l_y \times l_z$. Let us denote $V_{\text{non}}(P)$ as the set of these small voxels that contain at least one point in the point cloud $P$. Then, we randomly subsample $k$ voxels (without replacement) from the set $V_{\text{non}}(P)$, and retain all the points in $P$ that are contained within each of these chosen voxels. Following this randomized sampling strategy, we create multiple subsampled point clouds (of size $k$) from $P$. Finally, we provide these subsampled point clouds as the input to the classifier $F$. The intuition behind our defense mechanism is that, when the number of adversarially affected small voxels is bounded, the majority of the subsampled point clouds do not include any adversarial clusters and thus the majority vote among their labels predicted by $F$ may still correctly predict the label of the original point cloud $P$. In the limit $l_x \to 0, l_y \to 0, l_z \to 0$, our method coincides with the PointGuard defense (Liu, Jia, and Gong 2021).

In the clustering attack model, an adversary adds a limited number (1-3) of adversarial shapes to the original point cloud $P$ such that it is misclassified by the point cloud classifier $F$. Let $V_{\text{att}}(P)$ be the subset of voxels in $V_{\text{non}}(P)$ that are affected by the clustering attack. We consider a setting where the number of affected voxels is very small, i.e, $|V_{\text{att}}(P)| \ll |V_{\text{non}}(P)|$. However, in comparison to (Liu, Jia, and Gong 2021), we do not limit the number of points added/affected by the adversary within a cluster. Thus, the notion of certified perturbation size defined in (Liu, Jia, and Gong 2021) is not applicable under the clustering attack model. Note that typically an adversarial cluster is very dense, i.e., the number of points in the adversarial cluster is comparable to the total number of points in the point cloud.

Our defense mechanism is developed based on the derandomized smoothing technique from (Levine and Feizi 2020). Given an input point cloud $P$, let $S_k(P)$ be the set of all distinct randomly-ablated versions of $P$ according to the above described sampling strategy. For the base classifier $F$, a smoothed classifier $g$ is defined as:

$$g(P) = \underset{y \in \{1,2,\dots,c\}}{\arg\max} \ n_y(P) ,$$

where

$$n_y(P) = \sum_{P' \in S_k(P)} \mathbb{I}\{F(P') = y\}, \quad \forall y \in \{1, 2, \dots, c\}$$

denotes the number of point cloud ablations that were classified as class $y$. Note that the resulting smoothed classifier returns the most frequent prediction of the base classifier over the ablation set $S_k(P)$. We refer to the fraction of (test) point clouds that the smoothed classifier correctly classifies as standard/empirical accuracy.

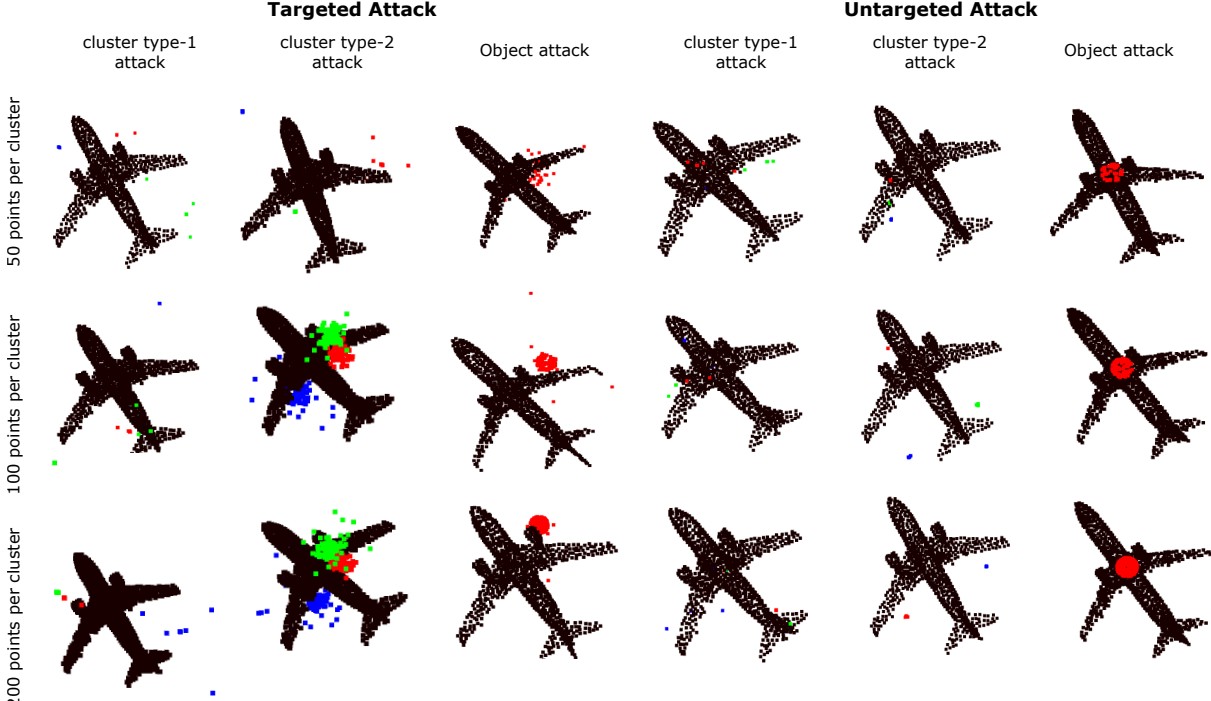

Figure 2: Visualization for adding adversarial clusters/objects for both targeted and untargeted attacks on ModelNet40. Here, type-1 and type-2 denote the two parameter settings, and the adversarial points are shown in red/green/blue color.

**Robustness Certificate** Here, we derive the certified robustness guarantee of our proposed defense mechanism. When performing derandomized smoothing, we classify all the point clouds in the ablation set $S_k(P)$ using the base classifier $F^1$. Of these classifications, at least $\binom{|V_{\text{non}}(P)|-|V_{\text{att}}(P)|}{k}$ will use none of the at most $|V_{\text{non}}(P)|$ voxels which may be affected by the adversary. Therefore, the number of classifications that may be affected by the adversarial clustering attack is at most:

$$\Delta(P) = \binom{|V_{\text{non}}(P)|}{k} - \binom{|V_{\text{non}}(P)| - |V_{\text{att}}(P)|}{k}.$$

Since the adversary can only alter the output of $\Delta(P)$ of the evaluations of the base classifier $F$, we obtain the following robustness guarantee for the smoothed classifier $g$ (Liu, Jia, and Gong 2021):

**Theorem 1.** *For any input point cloud $P$, and base classifier $F$, if:*

$$n_y(P) \geq \max_{y' \neq y} n_{y'}(P) + 2 \cdot \Delta(P),$$

*then for any point cloud $P'$ which differs from $P$ only in $|V_{\text{att}}(P)|$ voxels, $g(P') = y$.*

When the condition in the above theorem is met, the most frequent class (prediction of the smoothed classifier) is guaranteed to not change even if an adversarial cluster compromises every ablation (in $S_k(P)$) it intersects.

---

[1]Similar to (Liu, Jia, and Gong 2021), we train the base classifier $F$ on subsampled point clouds instead of the original point clouds.

**Remark 1.** *For practical purposes, we introduce two simplifications to above derandomized smoothed classifier: (i) we only consider a random subset (of size $N$) of the ablation set $S_k(P)$, (ii) within each voxel, we only randomly retain at most $m$ point cloud points.*

## 4 Experiments

### 4.1 Experimental Setup

**Datasets and Models** We conducted experiments on ModelNet40 (Wu et al. 2015) and ScanObjectNN (Dai et al. 2017) datasets. The ModelNet40 dataset is constructed using 3D CAD models, and each point cloud comprises of 1024 points and belongs to one of 40 different object categories. The standard split of 9,843 point clouds for training and 2,468 point clouds for testing is used. The ScanObjectNN dataset consists of 14,298 point clouds obtained by scanning real objects in indoor environments. The objects are categorized into 15 classes, where 11,416 objects are used for training and 2,882 for testing. Similarly to ModelNet40, 1024 points are used per point cloud. For our approach and the compared methods, we use the PointNet (Qi et al. 2017a) model as the base classifier, and the publicly available code is used[2].

**Evaluation Protocol** We follow the evaluation protocol of the recent PointGuard (Liu, Jia, and Gong 2021) method and compare against it on the clustering attacks discussed in Section 3.2. A vanilla PointNet classifier trained on the respective

---

[2]https://github.com/charlesq34/pointnet

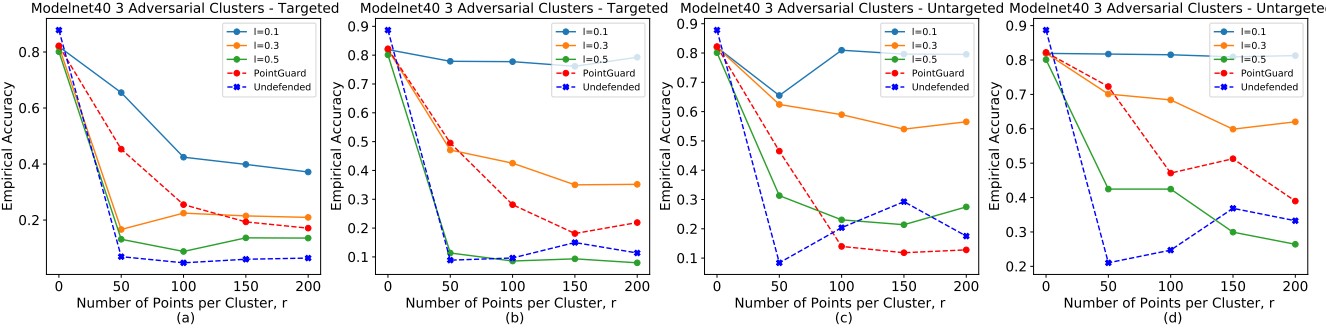

Figure 3: Targeted and untargeted clustering attacks on ModelNet40 with two attack parameter settings. Our defense model voxel size $l = 0.1$ significantly outperforms all other methods in all the cases.

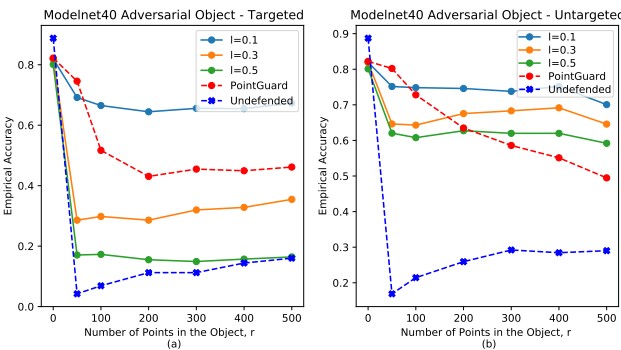

Figure 4: Targeted and untargeted object addition attacks on ModelNet40 point cloud dataset. In $r < 75$ regime, the performance of our model ($l = 0.1$) and PointGuard are roughly similar and when $r$ increases our model maintains the accuracy while PointGuard suffers.

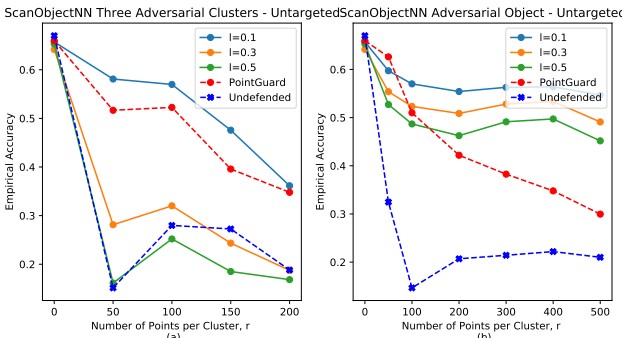

Figure 5: Untargeted attacks on ScanObjectNN point cloud dataset where the number of attack clusters is set to 3. Our model with voxel size $l = 0.1$ outperforms the other methods by a large margin. In $r = 200$ case both our model and PointGuard performs similarly, this might be due to large attack cluster radius.

dataset is used to generate the attack point clouds in the aforementioned four attack models: targeted and untargeted attack models with free-form clusters and specific objects, where the number of adversarial points and the number of adversarial clusters/objects are user-specified (see Figure 2).

We use the publicly available code[3] for attack generation with objective functions described in Section 3.2. We generate two types of adversarial cluster attacks for ModelNet40 with two different parameter settings. Specifically, $\mu_1, \mu_2, \mu_3$ parameters are set to $5, 10, 0.05$ and $0, 5, 0.05$. For ScanObjectNN the parameters are set to $50, 100, 0.5$. For the adversarial object attack, we set $\mu_1, \mu_2$ to $5, 1$ for both the datasets. Note that while these parameters are tuned to obtain attack clusters with minimal perturbation to the original point clouds, no hard constraint is enforced on the attack cluster radius.

For defense mechanisms, a base classifier is trained using sub-sampled point clouds rather than the original training set. Specifically, each point cloud is first sub-sampled to have $k \ll 1024$ number of points and passed to the model to opti-

---

[3]https://github.com/xiangchong1/3d-adv-pc

mize the loss. As discussed in Section 3.3, a model is evaluated by majority voting of the predictions on $N$ sub-sampled point clouds each with $k$ points. The main difference between our method and PointGuard is the sub-sampling strategy used, which in our case is voxel-based, where PointGuard simply uses random sampling of points. In the experiments, we also compare the baseline undefended classifier which is the vanilla PointNet classifier evaluated on the adversarial point clouds generated on the test set.

Unless otherwise specified, we used the default parameter setting of PointGuard. Specifically, for both ModelNet40 and ScanObjectNN datasets, we set $N = 10,000$ and $k = 16$. For our method, we experiment with different voxel sizes $l = l_x = l_y = l_z$ and the models are trained and tested on a Tesla P100 GPU.

**Evaluation Metric**   We use the empirical accuracy (the percentage of adversarial point clouds that are correctly classified) as the metric to evaluate and compare different defense mechanisms. Even though PointGuard uses certified accuracy, it does not apply to our method as the robustness

certificate measurement is different. In particular, PointGuard measures a radius of certified robustness, whereas our method provides certified robustness on the number of adversarial voxels/points.

## 4.2 Results

We evaluate both the attack variants (adversarial clusters and adversarial objects) on targeted and untargeted settings in ModelNet40, and since untargeted attacks are stronger, we evaluated them on ScanObjectNN as well. We report results for our method in three different voxel sizes, specifically, $l = 0.1, 0.3$, and $0.5$, and the $0.1$ version consistently outperforms. Some visualizations of the attack clusters are illustrated in Figure 2. In summary, our method outperforms Point-Guard and the undefended classifier in all the cases, especially when the number of adversarial points grows. The significance of our method is in the regime where the number of attack points is high but concentrated on sparse locations (typical clustering attack). We discuss the results in detail below:

**Adversarial Cluster Attack** In Figure 3 the empirical accuracy as a function of number of adversarial points ($r$) is reported for targeted and untargeted cluster addition attacks with two different parameter settings. For untargeted cluster addition attacks on ScanObjectNN are reported in Figure 5-(a). We limit the number of attack clusters to 3 for both datasets. In all six cases, our method with $l = 0.1$ produces significantly better empirical accuracies than PointGuard (almost $70\%$ better in some cases). Our model accuracy remains constant after $r = 100$ for ModelNet40 dataset while the ScanObjectNN accuracy decreases. The empirical accuracy of the undefended classifier quickly drops to $10 - 20\%$ while the empirical accuracy of our method is still much higher as $r$ increases.

**Adversarial Object Attack** We also compare our method with adversarial objects addition, where we add a single ball to the original point cloud. We evaluated targeted and untargeted attacks on ModelNet40 (Figure 4) and untargeted attack on ScanObjectNN(Figure 5-b). For both the attacks, our model maintains high empirical accuracy when voxel size $l = 0.1$ and it remains constant at $75\%$ when the number of points in the ball increases while the PointGuard saturates at $45\%$. Further, when the number of points is large, empirical accuracy measured with $l = 0.3, 0.5$ provides better results compared to PointGuard. This convincingly verifies the effectiveness of our defense mechanism with point addition attack.

## 5 Conclusions

In this paper, we have introduced a defense mechanism for strong clustering attacks on 3D point clouds based on a randomized smoothing strategy. Our defense mechanism is simple, yet effective and our experiments on synthetic and real point cloud datasets demonstrate the efficacy. We believe adversarial robustness on point clouds and other graph-structured data is an important research area, and we intend to investigate better defense mechanisms for attacks on such domains.

## Acknowledgments

Adrian Weller acknowledges support from a Turing AI Fellowship under grant EP/V025379/1, The Alan Turing Institute, and the Leverhulme Trust via CFI.

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
