# OpenReview forum: "Provable Defense Against Clustering Attacks on 3D Point Clouds"
_AAAI.org/2022/Workshop/AdvML — AAAI-22 AdvML Workshop LongPaper_

### Official Review · Reviewer_VaXN · 2021-11-27
**Review of the submission**

**Rating:** 7
**Confidence:** 4

**Review:**

This paper proposes a provable defense based on randomized smoothing for certifying robustness against clustering attacks on 3D point clouds.

Strengths:
- The motivation of using the random sampling strategy is clear.
- The theoretical results are provided.
- Some experiments can demonstrate the effectiveness of the method.

I also have some concerns about this work:
- Is it the first work to study provable defenses of 3D classifiers? If not, please discuss with previous work.
- Can you show the certified robustness in experiments?

Overall, I think this work is interesting and makes a contribution to the field.

---

### Official Review · Reviewer_29so · 2021-11-27
**An effective voxel-drop-based certified defense method on PointNet.**

**Rating:** 7
**Confidence:** 4

**Review:**

Pros:

1, The math is good, and the equations are clear.

2, The intuition is direct.

3, The paper is easy to follow.

Cons:

1, The number of comparison methods are not enough.

2, It will be better if a pipeline figure is added into the paper, which shows how the defense works.

---

### Decision · Program_Chairs · 2021-12-01

**Decision:**

Accept (Long Paper)

**Comment:**

Both reviewers agree to accept this paper. Please address their comments in the final version.